# Food and family care during the COVID-19 pandemic: A study of women's domestic workload during the first wave in Chile

Nathalie Llanos[1][☯], Lorena Iglesias[1][☯], Patricia Gálvez Espinoza[1][☯]*, Carla Cuevas[2][‡], Dérgica Sanhueza[2][‡]

1 Department of Nutrition, University of Chile, Santiago, Chile, 2 Project Fondecyt 11180370, University of Chile, Santiago, Chile

☯ These authors contributed equally to this work.
‡ CC and DS also contributed equally to this work.
* pagalvez@uchile.cl

**Data Availability Statement:** "All relevant data for this study are publicly available from the openICPSR repository (https://www.openicpsr.org/openicpsr/project/191341/version/V1).".

## Abstract

This study aimed to explore women's perceptions of domestic work related to food and family care during the first wave of the COVID-19 pandemic in Chile and its association with sociodemographic and health variables. We conducted a cross-sectional, analytical, non-probabilistic study. A sample of 2047 women answered an online self-report survey that included a Likert scale about the perception of domestic work associated with food. The survey also included an open comment section. The survey was available between May and June 2020, during the first wave of the COVID-19 pandemic and when most of the country had some degree of mobility restriction. 70.2% of participants perceived their domestic work as "regular"; being younger, having a higher educational level, caring for children or the elderly, and having worse self-perception of mental and general health status increased the chances of having a lower perception of the burden of these tasks. In comments, women declared how heavy the domestic work was, the challenges of being together with their families and of paid job requirements, and how family demands from them increased. Most women felt that their domestic work was heavier during this pandemic period: some groups of women could be at risk of being more affected by this extra workload at home. The importance of interventions and public policies with a gender perspective becomes relevant, considering the role of women in the home and the necessity to generate a social change regarding the domestic burden associated with gender.

## Introduction

The COVID-19 pandemic altered our lifestyles and family dynamics; it forced people to stay in their homes and work online; children and adolescents stayed at home without attending educational establishments in person. Unemployment also increased, generating economic instability for thousands of families [1, 2]. In this new context, the domestic environment becomes relevant, a space where, in general, the woman has a fundamental role.

**Funding:** This work was supported by the Ines Género at the University of Chile (no grant number-PGE and LIV) and the National Fund for Scientific and Technological Development (Fondecyt) program at the National Agency for Research and Development (ANID) (grant number: 11180370) (PGE). The funders had no role in the study design, data collection, analysis, publication decision, or manuscript preparation. No additional external funding was received for this study.

**Competing interests:** The authors have declared that no competing interests exist.

During the industrial revolution, paid work began to be carried out outside the home, and the social division of "work" and "family" arose, where the home became the space for leisure and resting from paid work [3] Here, the sexual division of labor was also generated with specific gender roles; men were part of the public world (labor market), and women were part of the private world, meaning the home [3]. For decades, it has been women who have been most associated with domestic work. The International Labor Office (ILO) has reported that women perform three times more unpaid domestic work than men worldwide [4], including caring for the family, managing household chores, and feeding the family [5, 6]. This would mean that women are in charge of modulating one of the essential food environments, the home food environment [7, 8].

The sanitary crisis due to COVID-19 pandemic had gender dimensions, a point repeatedly demonstrated in aspects such as the probability of becoming ill, mortality, and job losses [9–12]. Added to this, many women postponed their professional careers and personal well-being for the care of others [10]. Emerging research suggests that the crisis led to a dramatic increase in the double workload [13, 14]; women must fulfill both, their paid and unpaid work at home [15]. It has been described that the time spent by women in domestic work was significantly higher during the pandemic compared to a period without a pandemic, raising the average time spent doing these tasks by more than three and a half hours a day [16]. This also increases the risk of causing deterioration in women's physical and mental health [17–19]. Data showed that women had a higher level of stress than men [20], and compared to the pre-pandemic period, during the pandemic, they could increase their BMI and blood pressure [21]. This way, the pandemic crisis and women's responsibilities could also affect their general wellbeing.

In conclusion the pandemic has not only been a health crisis but also caused a crisis in the domestic space and home food environment, creating an ideal situation to make structural inequalities more visible [22–24].

In Chile, the scenario was not different. Before the pandemic, the first national economic evaluation of unpaid domestic and care work study in 2019 showed that of the total annual hours devoted to this work, about 71% is done by women [25]. However, this is usually under-valued. If unpaid domestic and care work were economically valued in the country, this would be equivalent to 22% of the Expanded Gross Domestic Product [25]. Additionally, concerning the home food environment, it has been observed that even though some women work outside the home, they are in charge of preparing the family meal and making decisions based on the family tastes and preferences [26, 27]. Evidence suggests that during the pandemic period, the household workload for women has increased [28]; 22.2% of Chilean women increased the time for doing household tasks by more than eight hours [29].

Despite women's relevance in the home food environment, the literature about women's specific tasks related to "feeding the family" during the pandemic is still scarce in Chile and Latin America. In this context, it is interesting to study how women perceived domestic work and the changes in their home food environment caused by the pandemic; these aspects have not been studied well in the country or Latin America until now. To fill this gap, we aimed (a) to explore women's perceptions of domestic work related to food and family care during the first wave of the COVID-19 pandemic and (b) to describe the association of domestic work with sociodemographic and health variables.

## Materials and methods

### Study design

We conducted an analytical, non-probabilistic cross-sectional study in 2020. Given the context of the health crisis and government measures to reduce the movement of people, we decided

to use digital methods to collect the information. According to the Organization for Economic Cooperation and Development, 87.4% of Chilean households have internet access [30]. The Ethics Committee (IRB) at the Faculty of Medicine of the University of Chile (Certificate # 044–2020) approved the study protocol. We obtained online consent from all participants in the study. Before the survey was displayed, a consent form was shown to all participants, explaining the study. At the end of this form was a question about whether they wanted to participate in the study. If they accepted to participate in the study, they marked yes, and then the survey was displayed. A thankful message was displayed if the answer was negative and the survey ended.

## Participants

Adult women between the ages of 25 and 65, who lived in Chile and had internet access, were invited to participate through social media platforms such as Facebook, Instagram, and WhatsApp. A total of 2,272 women responded to the survey. After reviewing the database and excluding women who did not reside in Chile and were not in the age range, a final database with 2,047 women was used. No incentive was given to any participant.

## Data collection

Following standardized methodologies [31, 32], we developed an online self-report survey that included closed and open questions based on the literature on domestic workload and experiences of women whom the research team worked with during the pandemic. The preliminary survey version was reviewed by four experts (two from the social science field and two from the health science field). The experts reviewed the language and cultural appropriateness. After this revision, we created a final survey version.

The final survey version had three sections. The main section included a Likert scale created by the research team based on women's experiences reported through several channels (previous research, media, and social networks, among others) and literature on the subject. The scale had 35 items and included three dimensions related to domestic work associated with food: spending on food and food purchases (8 items), food preparation (13 items), and family care (children, partners, and others) (14 items) (items are presented in Table in S1 Table). Each item had the possibility of 5 responses, from strongly agree to strongly disagree; we also included the option "no response" in case the woman perceived that the item did not apply to her. When any of the items were not answered or the option "no response" was chosen, the item and the survey were not considered, leading to a decrease in the total number of validated surveys used in the analysis (final total sample n = 1,425). The scale showed strong construct validity (KMO = .933) and internal consistency (Cronbach Alfa = .909).

In addition, the survey included demographic and socioeconomic variables such as age, level of education (incomplete high school education or completed high school, complete technical or incomplete technical or college degree, complete college degree, and incomplete or complete graduate degree), per capita monthly income (divided into quartiles into income equal to or less than $ 340, between $ 340 and $ 650, between $ 650 and $ 1,000, and income higher than $ 1,000), type of household (households without children, two-parent households with children, and single-parent households with children), number of people in the household, care of children or the elderly (yes or no), phase of confinement (according to the mobility restriction determined by the Ministry of Health: Total quarantine-Phase 1, Transition stage -Phase 2 with greater mobility during the week, but only essential services can be accessed on the weekend, and Opening stage-Phase 3 with more freedom on weekdays and weekends. We also included health variables such as self-perception of stress during the last

month (yes, no, maybe), self-perception of general health status (excellent, good, regular, and bad), and self-perception health status during the pandemic (no changes, worst, better).

Finally, the survey has an open-answer question that allowed women to add any comments related to the study topic. We received 550 valid comments from women in this questionnaire section.

The survey was available during May and June 2020, when most neighborhoods were confined.

## Data analysis

Each item on the Likert scale scores 1 to 5 depending on the item's directionality (positive, negative). The maximum score for the survey was 175 points. This score was divided into terciles, thus obtaining three categories: bad (<87), regular (88–107), and good (> 107). These categories respond to women's perception of domestic work related to food and family care during the pandemic. Descriptive statistics, percentages, means and medians, and standard deviation (SD) or medians and interquartile ranges (IQR) (depending on the distribution of the numerical variables) were used for the description of the study variables.

The association between women's perception of domestic work during the pandemic and sociodemographic and health factors were analyzed using univariate logistic regression in the first instance. Then, we used a multivariate logistic regression model with those sociodemographic factors with a statistically significant association. The strength of the association was evaluated using the Odds Ratio (OR) and its 95% confidence interval (95% CI). Significant values were associated with a two-tailed p-value less than 0.05. The STATA version 13 software was used to carry out the analyses.

We conducted a thematic analysis of the comments [33]. First, two team members read all comments to familiarize themselves with the data. Then, the researchers coded each comment using an inductive approach [34]. When necessary, we applied one or more codes to each comment. Then, the codes were grouped into categories according to similitudes. These categories were organized to obtain the themes [33]. We described these themes using quotes from the women's comments. We identified the origin of the quotes with a number and women's age.

## Results

### General sample characteristics

The median women's age was 36 years (IQR: 31–44). Almost half of the women (47.5%) had completed college studies, while 2.8% had an incomplete or complete high school education. The median per capita income was $ 650 (IQR: $ 340–$ 1,000). Women lived in a house with a median of 3 people (IQR: 2–4). The most prevalent type of household was two-parent with children (48.1%); more than half of the women surveyed had a child or an older adult under their care (61.3%); 116 (8.1%) women cared for a child and an older adult. 85.9% of women felt stressed during the last month (pandemic period); 57% self-perceived their general health status as "good." Regarding whether their self-perceived health had changed during the pandemic, 57.8% believed their health had not changed, and 36.3% indicated that their health status had worsened. Table 1 summarizes the characteristics of the sample.

### General perception of domestic work

The average score for the spending on food and food purchases dimension was 21 (SD: 7.48) points. In the food preparation dimension, the average score reached 37.4 (SD: 8.9) points, while for family care dimension, it was 39.6 (SD: 10.3) points. Women obtained an overall

**Table 1. Characterization of the sample by demographic, socioeconomic and health variables.**

| Variable | N (%) |
|---|---|
| *Level of education* | |
| Some high or complete high school | 40 (2.8) |
| Incomplete technical or college degree | 203 (14.4) |
| Complete college degree | 668 (47.5) |
| Postgraduate degree | 497 (35.3) |
| *Per capita income (dollars)* | |
| ≤ $340 | 355 (24.9) |
| > $340 y ≤ $650 | 364 (25.5) |
| > $650 y < $1.000 | 257 (18.0) |
| ≥ $1.000 | 449 (31.5) |
| *Type of home* | |
| Home without children | 545 (38.3) |
| Two-parent home with children | 685 (48.1) |
| Single-parent home with children | 193 (13.6) |
| *Number of people in the household* | |
| 1 | 52 (3.6) |
| 2 | 378 (26.5) |
| 3 | 374 (26.3) |
| 4 | 363 (25.5) |
| 5 o more | 258 (18.1) |
| *Caring for children or older adults* | |
| Yes | 873 (61.3) |
| No | 552 (38.7) |
| *Confinement phase* | |
| Phase 1 (Quarantine) | 613 (43.0) |
| Phase 2 (Transition) | 722 (50.7) |
| Phase 3 (Opening) | 90 (6.3) |
| *Stress* | |
| No | 55 (3.9) |
| Yes | 1224 (85.9) |
| Maybe | 146 (10.2) |
| *General health status* | |
| Excellent | 83 (5.8) |
| Good | 812 (57.0) |
| Poor | 482 (33.8) |
| Bad | 48 (3.4) |
| *Changes in health status in pandemic* | |
| Better | 84 (5.9) |
| No changes | 823 (57.8) |
| Worst | 518 (36.3) |

survey score of 93.1 (SD: 18.4) points, ranging from a minimum of 50 to a maximum of 107. None of the 1.425 women who completed the entire survey perceived their domestic work associated with food and family care during confinement as *good*. Most women (70.2%) considered their domestic workload to be *regular*. The items most negatively perceived by women were related to an increased need for cooking and spending more time in the kitchen, followed

by difficulties purchasing and accessing food. Moreover, concerns regarding family nutrition, meal routines, and healthy eating habits were also notably negative compared to other items (for detailed average scores for each item, refer to the supplementary material).

## Domestic work and sociodemographic variables

In the univariate analysis, we found a statistically significant association between a poorer perception of housework and age, educational level (just the postgraduate degree category was significant), the type of household, caring for children or the elderly, and the number of people in the household (just the categories of 3 and 4 members in families was significant). No association with per capita income was found.

The results from the multivariate analysis are shown in Table 2. We observed that age was associated negatively with the women's perceptions of the housework (OR 0.97 95% CI 0.96–0.99), meaning as age increases by one year, the probability of having a bad perception of the housework decreases by 3%. Regarding education, we found that having a postgraduate degree increases the probability of having a bad perception of domestic work by almost three times when compared to women who had complete or incomplete high school education (OR 3.14 95% CI 1.24–7.96). We observed that women living in single-parent or two-parent households with children increase the probability of having a lower perception of domestic work by more than 95% (OR: 2.15 CI95% 1.28–3.59 and OR: 1.95 95% CI 1.24–3.04, respectively), in comparison with those women living in households with no children.

Concerning the type of confinement, we found no significant association between quarantine (phase 1) and transition (phase 2). However, being in the initial opening phase and having more freedom (phase 3) decrease the probability of having a bad perception of domestic work by half (OR: 0.41 95% CI 0.21–0.79). Being in charge of children or an elderly person also increases the probability of women having a bad perception of housework during the pandemic by more than twice compared to not having this responsibility (OR: 1.92 95% CI 1.32–2.77). We did not find significant differences in the multivariate model regarding the number of people in the household. Finally, for the variables related to the women's perception of their health, the self-perception of stress was not significant in the multivariate model.

## Domestic work and health variables

For the health variables, the analysis showed statistical significance for feeling stressed during the last month, for all categories of perception of general health status, and for the perception that health has worsened during the pandemic (Table 2).

For the self-perception of general health status, the worse the perception of health was, the higher probability that women negatively perceived domestic work related to feeding and caring for the family. A similar situation occurred for the perception of their health status worsened during the pandemic (OR: 5.30 95% CI 2.74–10.22) (Table 2).

In the open-answer question, women added comments that allowed us to expand what we found in the Likert-type scale. Most women that commented described this pandemic as a "bittersweet period." Some of them enjoyed being at home more, having more time for themselves, and enjoying their families. However, at the same time, more time at home meant a more significant domestic workload. We identified three main themes about how women perceived this workload during the COVID-19 pandemic: 1) General perceptions about domestic workload, 2) Combining domestic work with paid work, and 3) Demands from their family members: Below is a detailed explanation of each of these themes.

**1) General perceptions about domestic workload.** Most women tended to agree that their household workload has increased, both in the time needed to get everything done and

**Table 2. Univariate and multivariate analysis of variables associated with the women's perception of domestic work related to food and family care during the COVID-19 pandemic period (n = 1406).**

| Variables | OR unadjusted | IC95% | P value | OR adjusted | IC95% | P value |
|---|---|---|---|---|---|---|
| Age | 0.99 | 0.98–1.00 | 0.050 | 0.97 | 0.96–0.99 | 0.005 |
| Confinement phase | | | | | | |
| Phase 1 (ref) | 1.00 | | | 1.00 | | |
| Phase 2 | 1.08 | 0.86–1.37 | 0.496 | 1.11 | 0.86–1.44 | 0.416 |
| Phase 3 | 0.43 | 0.24–0.79 | 0.006 | 0.41 | 0.21–0.79 | 0.008 |
| Level of education | | | | | | |
| Some high or complete high school (ref) | 1.00 | | | 1.00 | | |
| Incomplete technical or college degree | 2.16 | 0.86–5.42 | 0.101 | 2.25 | 0.86–5.91 | 0.099 |
| Complete college degree | 2.32 | 0.96–5.61 | 0.062 | 2.54 | 1.01–6.45 | 0.050 |
| Post graduate degree | 2.77 | 1.14–6.72 | 0.025 | 3.14 | 1.24–7.96 | 0.016 |
| Type of Home | | | | | | |
| Home without children (ref) | 1.00 | | | 1.00 | | |
| Two-parent home with children | 2.16 | 1.67–2.81 | <0.001 | 1.95 | 1.24–3.04 | 0.004 |
| Single-parent home with children | 2.13 | 1.48–3.05 | <0.001 | 2.15 | 1.28–3.59 | 0.004 |
| Caring for children or older adults | | | | | | |
| No (ref) | 1.00 | | | 1.00 | | |
| Yes | 2.55 | 1.98–3.29 | <0.001 | 1.92 | 1.32–2.77 | 0.001 |
| Number of people in the household | | | | | | |
| 1 (ref) | 1.00 | | | 1.00 | | |
| 2 | 1.33 | 0.64–2.76 | 0.441 | 1.16 | 0.51–2.61 | 0.715 |
| 3 | 2.13 | 1.04–4.39 | 0.04 | 0.82 | 0.35–1.93 | 0.645 |
| 4 | 2.15 | 1.04–4.44 | 0.038 | 0.89 | 0.38–2.13 | 0.803 |
| 5 | 1.72 | 0.82–3.61 | 0.15 | 0.76 | 0.32–1.82 | 0.534 |
| Per capita income (dollars) | | | | | | |
| ≤ $340 | 1.00 | | | | | |
| > $340 y ≤ $650 | 1.12 | 0.81–1.55 | 0.49 | | | |
| > $650 y < $1,000 | 1.39 | 0.98–1.96 | 0.063 | | | |
| ≥ $1,000 | 1.06 | 0.78–1.44 | 0.725 | | | |
| Stressed during last month | | | | | | |
| No (Ref) | 1.00 | | | 1.00 | | |
| Yes | 4.89 | 1.94–12.35 | 0.001 | 2.59 | 0.98–6.86 | 0.054 |
| Maybe | 1.41 | 0.50–3.99 | 0.522 | 1.09 | 0.37–3.24 | 0.875 |
| Self-perception of health status | | | | | | |
| Excellent (Ref) | 1.00 | | | 1.00 | | |
| Good | 8.54 | 2.67–27.35 | <0.001 | 6.88 | 2.08–22.72 | 0.002 |
| Poor | 19.23 | 5.99–61.78 | <0.001 | 10.51 | 3.14–35.25 | <0.001 |
| Bad | 24.53 | 6.79–88.61 | <0.001 | 11.82 | 3.09–45.16 | <0.001 |
| Changes in health status in pandemic | | | | | | |
| Better (Ref) | 1.00 | | | 1.00 | | |
| No changes | 1.91 | 0.99–3.68 | 0.053 | 1.80 | 0.91–3.58 | 0.091 |
| Worst | 5.30 | 2.74–10.22 | <0.001 | 3.59 | 1.79–7.21 | <0.001 |

OR: Odds Ratio. CI95%: Confidence Interval 95%. ref: Reference. Phase 1: Quarantine, full lockdown, strict restriction of mobility. Phase 2: Transition, strict restriction of mobility only on weekends. Phase 3: Opening, no restriction of mobility

the effort put into finishing each task; most of them agreed that housework activities have been "heavier" during this period.

"*During this period, housework has been more intense since we are all at home...*"

(P470, 45).

Among the household tasks, cooking was perceived as increasing the most. Cooking was described mainly as a female task, which burdened women heavily. Before the pandemic, women used to eat in restaurants or eat at their workplaces. Furthermore, some women mentioned that prior to the pandemic, they had a person who helped with house duties. However, they did not have this help during the pandemic due to the sanitary measures.

This extra work was paired with some women struggling with the changes posed by the COVID regulations and the changes in their daily life and routines. This new context was associated with women feeling that their physical and psychological well-being was being affected. Some women mentioned feeling "exhausted" or "stressed" due to overworking during the day.

"*I feel physical and emotional tiredness since I am the only one responsible for the housework*"

(P96, 40).

Additionally, women indicated that most of the housework relied on them. Some women felt like they were in a "disadvantaged" position concerning their involvement in housework. This thought made them often feel disappointed since they perceived housework was mainly for women in the house, and this was considered "less important" by other family members.

"*Unfortunately, housework is underestimated, it has no wages/salary, and it is not deemed important*".

(P1312, 40)

One group of women described how they were able to organize and did the housework together with their families during the pandemic. However, it was the women who organized the different labors in the house. Some women saw household tasks as something that had to be done by the family as teamwork.

"*...I have learned to delegate tasks to all family members... according to their age... in order for them to see the importance of teamwork and that in extreme situations like this one [COVID-19 pandemic], the collaboration of everyone is needed.*"

(P148, 37).

Some women even indicated that the pandemic has been useful in getting their partners more involved in domestic work.

**2) Combining domestic work with paid work.** Even when some women indicated reaching an equilibrium between domestic work and paid work, most women working during the pandemic reflected on how complicated it has been trying to comply with both jobs. Because they had this double burden of work, some indicated changes in their routines and quality of life due to completing all tasks. Some expressed that they lost their housekeepers due to sanitary measures, which increased their housework load.

"*Sleep time is altered because there is more to do during the day and therefore, at night you work in your paid job, especially if you live with very elderly parents; one independent and the other semi-dependent.*"

(P194, 56)

Women also mentioned that trying to accomplish paid jobs and unpaid household tasks added extra stress to their lives, especially at the beginning of the lockdown period when they were trying to adapt to being at home all day and all together. In addition, some women pointed out the difficulties of concentrating in the home when their other family members, such as their children, always asked them for stuff.

"*Even though we are two adults in the house, I have had to do many things related to the home and the care of my daughter, which does not give me a space to concentrate and focus on my work. I am always doing everything at the same time.*"

(P465, 34)

Finally, some women mentioned feeling that they worked more now for their paid jobs. They felt it was difficult to control their schedule and office hours because they had to do domestic work during the day, so they compensated for this time by extending their work hours.

"*I feel that I have more paid work than before. . . I feel that the rest think that being at home makes me available for more hours a day and that it has to be immediate.*"

(P620, 44)

**3) Demands from their family members.**   Women felt that some family members were demanding a lot from them during this period, especially their kids. According to women, another domestic demand that increased during the pandemic was caring for their children. This chore was perceived as worse when they had to care for their toddlers and work for their paid job.

"*. . .But, today paid work is also from home, and I end up doing it when the children sleep, they are under two years old, therefore paid work is from 9:00 p.m. to 2:00 or 3:00 a.m. and housework from 7:00–21:00. . .I sleep little at night, and sometimes I sleep when the children take naps.*"

(P69, 40)

Furthermore, as children had to do homeschooling, women highlighted that their school assignments demanded a lot of their time. Some women expressed that besides being a mom, they became teachers too during this period; sometimes, this made them feel stressed or anxious. This relatively new assignment was perceived worst in women with paid jobs.

"*I am overwhelmed with schoolwork, between Zoom classes, the activities they send and the time I spend doing the activities with my little son (2nd grade). . . I don't have time to do anything*"

(P640, 44)

"*I am at times very overwhelmed since I have put aside the academic issue of my children trying to balance everything else and trying to maintain my mental health and theirs in acceptable terms*"

(P250, 37)

Women also described how the elderly relatives in their families added to their workload. In some cases, these elderly relatives were not living with the women; however, they were a constant concern for women due to isolation.

"*Regarding my parents, they're elderly with risk factors, I do feel that I should support them by supplying food to maintain a healthy diet without them leaving the home.*"

(P332, 29)

## Discussion

This study explored women's perceptions of domestic work related to feeding and caring for the family during confinement due to COVID-19 and how these perceptions were related to sociodemographic and health variables. Child care, food preparation, and other tasks were perceived even worse for some women during this health crisis. This study contributes with supporting evidence about the role and the inequalities women had during the pandemic related to specific household tasks such as food purchases, food preparation, and family care that should be considered in futures similar sanitary and social crises.

Most women generally perceived domestic work's burden as "regular." In concordance, women also expressed in the open-answer question that this period had advantages and disadvantages. It was a challenging period where they had to balance paid work with unpaid work. These general results align with previously published data showing negative perceptions about women's double burden during this health crisis [35, 36]. Additionally, research demonstrated that men also perceived an increase in their domestic workload, however, this extra work is considered a slight increase [37, 38] and lower than what women do [39, 40]. Some research has indicated that the pandemic could produce "regressive shifts in gender role attitudes" (p. 30) [41].

Regarding sociodemographic variables, we found that women in the older groups had a better perception of domestic work related to food and family care during the pandemic than younger women. Although domestic and care tasks occupy an essential part of a woman's time, new generations have had a cultural and social openness that distances them from the domestic space [42]. Today, there is a questioning of gender roles; new generations demand that domestic tasks be shared with their partners, especially when women enter the job market and share the role of provider. This situation is very different from the one experienced by older generations [43].

Furthermore, we observed a trend between having a higher educational level and a higher probability of having a poor perception of the household burden, especially significant for women with postgraduate degrees. This perception could be because most jobs that were lost were in the service, formal and informal trade, and industrial sectors [44]. In Chile, these sectors are the primary sources of employment for women with lower educational levels [45]. Therefore, it is possible that women with a higher socioeconomic level did not lose their jobs but had to reconcile paid work with unpaid work at home, making them feel more overwhelmed than their counterparts, as they pointed out in some of the comments. Finally, it is

also relevant to consider that they could not count on the support of domestic workers due to the confinement and mobility restrictions.

When we analyzed women caring for children or the elderly, we found that the probability of perceiving domestic labor as bad was almost double that of those who did not have this responsibility. This finding is in concordance with other studies on the topic carried out during the pandemic in which the presence of children or the elderly in the home worsened the perception of the domestic workload, especially when children are homeschooled or elderly people have special care needs [12, 23, 46]. In concordance with this, women living in households with children had more than a 90% probability of having a bad perception of domestic work than women living without children. This fact is in line with women indicating that they had a higher demand from their children, even becoming mothers and teachers simultaneously during the pandemic. However, we did not analyze the children´s ages to evaluate the differences in the perception of burden in domestic work. Studies before the pandemic show that working mothers of preschool or elementary school children perceive more domestic workload [47, 48].

In addition, we found almost the same probability of having a lower perception of domestic labor in women living with partners and children and those living without a partner but with children. Having a partner in the home does not appear to be perceived as an extra domestic burden because women are used to doing this extra household work, so it wasn't perceived as heavier during the pandemic. This finding is surprising because the evidence shows that women spend 9 hours a week more than men cooking and cleaning, 14 hours more than men caring for children, and 57% of men dedicated 0 hours a week to this activity [49]. Similar results were found in studies conducted in Australian and Canadian couples, where the time spent caring for children was much higher during confinement, with mothers spending more time than fathers, even though fathers increased the time they take care of their children [38, 50]. A recent study in Argentina which also used an online survey to collect the data, showed that women assumed a more significant proportion of domestic tasks and childcare. The authors concluded that one of the reasons the gender gap increased was women, besides the household work, continued working from home [51]. It is necessary to note that we did not collect information about the composition by sex of the two-parent households; therefore, we assumed that most of our women were part of households with a man and a woman, as most of the families in the country [52].

As expected, women in phase 3 of the health crisis plan, meaning more freedom, were more likely to perceive domestic labor better than their counterparts. Being in phase 3 meant freedom of mobility on weekdays and weekends, the opening of some schools with voluntary attendance, the opening of gyms and restaurant terraces, and social gatherings with limited capacity in residential spaces were allowed [53], as such, women had the opportunity to spend time doing other activities that they enjoyed). Domestic workers were also able to resume working for some women. In addition, with more freedom on weekends [54], women living in zones in phase 3 could use these days as a time of distraction from work and homeschooling.

Concerning women's health status, in our sample, a high percentage of participants felt stressed, and more than a third felt their health status was not good. These findings align with several studies conducted during the pandemic in quarantine periods, showing that people experienced the worst level of health, especially women. The lower access to medical check-ups, the changes in diet due to confinement and lower access to healthy food, and the high sedentary lifestyle and low level of physical activity could explain this worst level of health in women [21, 55]. In our study, a negative perception of mental and general health status was also associated with a worse perception of domestic work. The fact that most women do those tasks related to caring for others [5, 6], but in sanitary crisis, with the school closure and the mobility restriction, these tasks were done every day and most of the day, and at the same

time, most of them had to comply with paid job requirements, all could cause more pressure on women. In our study, women commented that during the pandemic, they felt stressed and domestic work made them feel stressed and overwhelmed. This increase in the double burden of household work, added to the families economic straits, could lead to a higher stress level [56]. Furthermore, literature on the topic has indicated that women, especially those with school-age children, felt stressed and worried about the pandemic consequences, altering their mental health [57] which our participants also pointed out. Having women with less time for themselves, less leisure time, and less communication with others close to her could lead to physical and emotional problems and even depression and anxiety. If, in this context, it is added sanitary conditions such as the mobility restriction, in which women could not access health care services, and the increase in domestic violence, in some cases, it is easier to visualize why there is a deterioration of women's health status.

The present study highlights the potential effect of a health crisis like the COVID-19 pandemic on a social structure as the family. This study mainly shows the effect on women and how they perceived the domestic workload, a role inside the family traditionally given to them by society. Some participants had to overcome the challenges of reconciling paid and unpaid work during a stressful period. This context made women perceive the domestic workload as regular and bad during the lockdown. Policymakers must create more social and health policies with a gender approach that allows women to face the crisis, the post-crisis period and be prepared for new crisis. An excellent example of promoting a gender approach is found in the Food Guidelines. Some guidelines surpass mere recommendations for food intake and incorporate social and cultural aspects. For instance, the latest Chilean Food Guidelines, published in 2022, have embraced a gender approach by incorporating the notion that 'household chores are everyone's responsibility' [58]. This concept should be promoted in schools and other contexts, including workplaces. Another pressing public health need is the establishment of mental health programs with a gender perspective within the public health system. Primary care settings have the potential to host these programs.

Furthermore, to alleviate stress during confinement or similar circumstances, governments should contemplate public policies about regulations on paid work, flexible or remote working hours, and clear guidelines to promote employees' emotional and mental well-being [59, 60]. Paid parental or family leave remains a crucial support for families in managing care responsibilities for children or other dependents [61]). Lastly, it has been recommended that governments implement tax policies favoring sectors where women are predominantly employed, thereby facilitating their (re)entry into the labor market [61]. All these policies could contribute to better integrating domestic and paid work.

Despite our valuable results, this study has structural limitations: First, we did not use a probabilistic sample, meaning there was selection bias; women who participated in this study had access to the internet and wanted to answer the survey; therefore, the results of this study could not be extrapolated to all Chilean women. Second, most of our participants had a higher level of education; consequently, we may not reach the most vulnerable population in which household tasks could be perceived differently from our sample. Third, we considered self-perception of mental and health status in our study through one question and not from previously validated instruments. Fourth, we collected data during the first wave of the COVID-19 pandemic in Chile, a stressful period filled with uncertainty; therefore, it is possible that women perceived their workload even worse because of the negative context. Finally, we did not have a baseline or data from men to compare our results to enrich our analysis. In future research, including a representative sample of men to understand their perspective and contribution to household responsibilities during a health crisis would be relevant to obtain a complete and balanced picture of how the crisis affected both genders.

## Conclusions

This study contributes to the literature by exploring the perception of Chilean women about the burden of domestic work related to feeding and caring for the family during the first wave of the COVID-19 pandemic in Chile. It complements the existing information about time spent by women on domestic tasks, adding that this is perceived as regular or bad. The probability of perceiving this domestic work as worse increased in women who took care of children or the elderly at home, in younger women, those with a higher level of education, single women or women living with partners and children, and those with worse perception of mental and physical status. Comments from participants indicated that the pandemic is a challenging time for taking care of the family and other domestic tasks. The evidence provides interesting data to raise awareness regarding the need to generate cultural changes regarding domestic tasks. Additionally, the data provided in the current study could work as a baseline for future research that evaluate domestic work in a post-pandemic period.

## Supporting information

**S1 Table. Items for the three scales and their descriptive statistic.**
(DOCX)

## Acknowledgments

The authors thank Mr. Brian Boyle and Ms. Danitza Osorio for helping with the English Edition of the manuscript.

## Author Contributions

**Conceptualization:** Nathalie Llanos, Lorena Iglesias, Patricia Gálvez Espinoza, Carla Cuevas, Dérgica Sanhueza.

**Formal analysis:** Nathalie Llanos, Lorena Iglesias, Patricia Gálvez Espinoza, Dérgica Sanhueza.

**Investigation:** Patricia Gálvez Espinoza, Carla Cuevas, Dérgica Sanhueza.

**Methodology:** Patricia Gálvez Espinoza, Carla Cuevas, Dérgica Sanhueza.

**Project administration:** Patricia Gálvez Espinoza, Carla Cuevas.

**Validation:** Dérgica Sanhueza.

**Writing – original draft:** Nathalie Llanos, Lorena Iglesias, Patricia Gálvez Espinoza.

**Writing – review & editing:** Nathalie Llanos, Lorena Iglesias, Patricia Gálvez Espinoza, Carla Cuevas, Dérgica Sanhueza.

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
