## [Decision Letter · Decision Letter 0]

13 Mar 2023

PONE-D-22-26228Food and family care during the COVID-19 pandemic: A study of women’s domestic workload during the first wave in ChilePLOS ONE

Dear Dr. Galvez Espinoza,

Thank you for submitting your manuscript to PLOS ONE. After careful consideration, we feel that it has merit but does not fully meet PLOS ONE’s publication criteria as it currently stands. Therefore, we invite you to submit a revised version of the manuscript that addresses the points raised during the review process.

We look forward to receiving your revised manuscript.

Kind regards,

Ali B. Mahmoud, Ph.D.

Academic Editor

PLOS ONE

Journal Requirements:

“This work was partially supported by the National Agency for Research and

Development (ANID) and its National Fund for Scientific and Technological Development (Fondecyt) program, (grant number: 11180370) (PGE), and by the Health Research Grant 2016, from the College of Medicine, Universidad de Chile (PGE and LIV).

Reviewers' comments:

Reviewer's Responses to Questions

**Comments to the Author**

1. Is the manuscript technically sound, and do the data support the conclusions?

Reviewer #1: Yes

Reviewer #2: Partly

2. Has the statistical analysis been performed appropriately and rigorously? 

Reviewer #1: Yes

Reviewer #2: N/A

3. Have the authors made all data underlying the findings in their manuscript fully available?

Reviewer #1: Yes

Reviewer #2: No

4. Is the manuscript presented in an intelligible fashion and written in standard English?

Reviewer #1: Yes

Reviewer #2: Yes

5. Review Comments to the Author

Reviewer #1: I have some (small) cooments that I think would allow to clarify some points:

1) Regarding previous literature, there are some papers worth mentioning. They are papers for LAC countries analyzing the impact of the pandemic on gender differences in labor outcomes and its connection with care work and gender difference in time use:

- Costoya, V., Echeverría, L., Edo, M., Rocha, A. y Thailinger, A. (2022). “Gender Gaps within Couples: Evidence of Time Re allocations during COVID 19 in Argentina.” Journal of Family and Economic Issues, 43: 213–226.

- Cucagna, E. y Romero, J. (2021). “The gendered impacts of COVID-19 on labor markets in Latin America and the Caribbean.” World Bank Gender Innovation Lab for Latin America and the Caribbean Policy Brief.

- Romero, T.H. y Reys, A. (2020). “Empobrecimiento de los hogares y cambios en el abastecimiento de alimentos por la COVID-19 en Lima, Perú.” Revista de Recursos en Internet sobre Geografía y Ciencias Sociales.

- Viollaz, M., Salazar-Saez, M., Flabbi, L., Bustelo, M. y Bosch, M. (2022). “The COVID-19 Pandemic in Latin American and Caribbean Countries: The Labor Supply Impact by Gender.” IZA Discussion Paper 15091.

2) Considering the previous point, the authors may want to stress more the fact that they are analyzing self-perceptions of domestic work, while previous literature considers time use information or whether women/men perform care activities.

2) It would be good to clarify whether the dependent variable in the regression analysis is defined as =1 if the score is "bad" or whether it is =1 if the score is "bad" or "regular".

Reviewer #2: While the manuscript presents insights into women's domestic workload related to food and family care during the first wave of the COVID-19 pandemic in Chile, there are some significant concerns that need to be addressed.

Firstly, the methodology used in the study is too simple and does not provide enough depth to draw meaningful conclusions. The study relies on a Likert scale and an open comment section, which may not be sufficient to capture the complexity of women's experiences during the pandemic adequately.

The authors developed their own scales but do not in detail describe the process, nor use proper statistical methods of testing and validating the items, and scales. Therefore, the manuscript lacks innovation in terms of the research methods used, which limits its impact.

Also, the study is qualitative, and as it has been three years since the COVID-19 pandemic began, it is unlikely to merit high-impact publication. While qualitative studies can provide valuable insights, they are generally considered less rigorous than quantitative studies and may not have the same impact as more innovative research.

At this stage there are hundreds of articles on COVID, why the study is submitted now if the data collection took place in 2020.

You may include some, such as to improve your study:

KABBOUT, R.E. & ZAITER, R. (2022). Covid-19 Related Stressors and Performance: The Case of Lebanese Employees During the Pandemic. International Journal of Management Science and Business Administration, 8(5), 14-25.

Sahni, D.J. (2020). Impact of COVID-19 on Employee Behavior: Stress and Coping Mechanism During WFH (Work From Home) Among Service Industry Employees. International Journal of Operations Management, 1(1), 35-48.

Impact of Social Media, Extended Parallel Process Model (EPPM) on the Intention to Stay at Home during the COVID-19 Pandemic

The authors could consider revising the research methods used, incorporating more quantitative data, and adding a more innovative angle to the study to make it more impactful. Alternatively, the authors could consider submitting the manuscript to a journal that is more focused on qualitative research.

6. PLOS authors have the option to publish the peer review history of their article (what does this mean?). If published, this will include your full peer review and any attached files.

Reviewer #1: No

Reviewer #2: No

---

## [Author Response · Author response to Decision Letter 0]

15 May 2023

We appreciate the reviewers for your precious time in reviewing our paper and providing valuable comments. We have carefully considered the comments and tried our best to address every one of them. 

Below, we provide the point-by-point responses. All modifications in the manuscript have been highlighted in red

I. Journal Requirements:

Comment #1: When submitting your revision, we need you to address these additional requirements. 

Response: Thank you for letting us know about these requirements. We have reviewed the style requirements and fixed the manuscript accordingly.

Comment #2: Please provide additional details regarding participant consent. In the ethics statement in the Methods and online submission information, please ensure that you have specified (1) whether consent was informed and (2) what type you obtained (for instance, written or verbal, and if verbal, how it was documented and witnessed). If your study included minors, state whether you obtained consent from parents or guardians. If the need for consent was waived by the ethics committee, please include this information. 

Response: Thank you for allowing us to clarify this point. All participants were older than 18 years old. We obtained online consent for participating in the study. Before the survey was displayed, a consent form was shown to all participants, explaining the study. At the end of this form was a question about whether they wanted to participate in the study. If they accepted to participate in the study, they marked yes, and then the survey was displayed. A thankful message was displayed if the answer was negative and the survey ended. This information was re-written to the new version of the manuscript.

Comment #3: If you are reporting a retrospective study of medical records or archived samples, please ensure that you have discussed whether all data were fully anonymized before you accessed them and/or whether the IRB or ethics committee waived the requirement for informed consent. If patients provided informed written consent to have data from their medical records used in research, please include this information. 

Response: Thank you for your comment. We did not use medical records or archived samples. 

Comment #4: Thank you for stating in your Funding Statement: 

“This work was partially supported by the National Agency for Research and

Development (ANID) and its National Fund for Scientific and Technological Development (Fondecyt) program, (grant number: 11180370) (PGE), and by the Health Research Grant 2016, from the College of Medicine, Universidad de Chile (PGE and LIV).

Response: Thank you for letting us amend our funding statement. We have rewritten it and added it to the cover letter. 

Comment #5: We note that the grant information you provided in the ‘Funding Information’ and ‘Financial Disclosure’ sections do not match.

Response: Thank you for letting us know about this mistake. We have fixed the funding information. 

II. Reviewers' comments

REVIEWER #1: 

I have some (small) comments that I think would allow to clarify some points: 

Comment #1. Regarding previous literature, there are some papers worth mentioning. They are papers for LAC countries analyzing the impact of the pandemic on gender differences in labor outcomes and its connection with care work and gender difference in time use:

- Costoya, V., Echeverría, L., Edo, M., Rocha, A. y Thailinger, A. (2022). “Gender Gaps within Couples: Evidence of Time Re allocations during COVID 19 in Argentina.” Journal of Family and Economic Issues, 43: 213–226.

- Cucagna, E. y Romero, J. (2021). “The gendered impacts of COVID-19 on labor markets in Latin America and the Caribbean.” World Bank Gender Innovation Lab for Latin America and the Caribbean Policy Brief.

- Romero, T.H. y Reys, A. (2020). “Empobrecimiento de los hogares y cambios en el abastecimiento de alimentos por la COVID-19 en Lima, Perú.” Revista de Recursos en Internet sobre Geografía y Ciencias Sociales.

- Viollaz, M., Salazar-Saez, M., Flabbi, L., Bustelo, M. y Bosch, M. (2022). “The COVID-19 Pandemic in Latin American and Caribbean Countries: The Labor Supply Impact by Gender.” IZA Discussion Paper 15091.

Response: Thank you very much for your recommendation. We have reviewed and incorporated these studies into the manuscript. Please, refer to the introduction and discussion section.

Comment #2 Considering the previous point, the authors may want to stress more the fact that they are analyzing self-perceptions of domestic work, while previous literature considers time use information or whether women/men perform care activities. 

Response: Thank you very much for the comment. In the discussion section, we have added a final paragraph where it is expressed as a limitation of the study that the data have been self-reported.

Comment #3. It would be good to clarify whether the dependent variable in the regression analysis is defined as =1 if the score is "bad" or whether it is =1 if the score is "bad" or "regular". 

Response: Thank you very much for allowing clarify this aspect. As the reviewer indicated, the dependent variable was the women's perception of domestic work related to food and family care during the COVID-19 pandemic. None participant perceived their domestic work as "good." Therefore, we defined 1 as the "bad" category in the regression analysis.

REVIEWER #2: 

While the manuscript presents insights into women's domestic workload related to food and family care during the first wave of the COVID-19 pandemic in Chile, there are some significant concerns that need to be addressed.

Comment #1. Firstly, the methodology used in the study is too simple and does not provide enough depth to draw meaningful conclusions. The study relies on a Likert scale and an open comment section, which may not be sufficient to capture the complexity of women's experiences during the pandemic adequately. 

Response: Thanks to the reviewer for allowing us to reflect on this point. We agree with the reviewer that this Likert scale collects perceptions about different topics, and this could overlook the problem that our study considered. However, it is a suitable methodology for collecting information during the pandemic time, considering the sanitary conditions people were living in (long lockdown period). Additionally, this methodology allowed participants to answer the questionnaire quickly (15 to 20 minutes), which was an advantage. This methodology was a good initial approach to what happened with domestic work and women during times of crisis..

Further studies, including complex methodologies, should be conducted to understand the phenomenon integrally. We have added this reflection to the discussion section. 

Comment #2. The authors developed their own scales but do not in detail describe the process, nor use proper statistical methods of testing and validating the items, and scales. Therefore, the manuscript lacks innovation in terms of the research methods used, which limits its impact. 

Response: We thank the reviewers for allowing us to reflect on this point. We have added a paragraph detailing the process of scale creation. Please, refer to the material and methods section. 

We recognized the methodology limitations and added them to the discussion section. However, this study gives a first approach to the context regarding domestic work women who lived during a sanitary crisis, which is an innovative situation per se. Further studies could use information from our study as a baseline.

Comment #3. Also, the study is qualitative, and as it has been three years since the COVID-19 pandemic began, it is unlikely to merit high-impact publication. While qualitative studies can provide valuable insights, they are generally considered less rigorous than quantitative studies and may not have the same impact as more innovative research. At this stage there are hundreds of articles on COVID, why the study is submitted now if the data collection took place in 2020. 

Response: Thanks to the reviewer for this comment. Qualitative and Quantitative studies can show different angles on a topic. For our research team, both approaches can give valuable information about a phenomenon. 

Even though the pandemic has been almost over in recent months, the results from this study could be a source of information for creating measures to protect women and families from future similar crises. For this reason, the study and its results merit publication. Additionally, we are contributing to the scarce evidence from Latin America. 

Comment #4. You may include some, such as to improve your study: 

KABBOUT, R.E. & ZAITER, R. (2022). Covid-19 Related Stressors and Performance: The Case of Lebanese Employees During the Pandemic. International Journal of Management Science and Business Administration, 8(5), 14-25.

Sahni, D.J. (2020). Impact of COVID-19 on Employee Behavior: Stress and Coping Mechanism During WFH (Work From Home) Among Service Industry Employees. International Journal of Operations Management, 1(1), 35-48.

Impact of Social Media, Extended Parallel Process Model (EPPM) on the Intention to Stay at Home during the COVID-19 Pandemic

Response: Thanks to the reviewer for these suggested manuscripts. Due to the relevance to our study topic, we have added two of them to the discussion section (Kabbout & Zaiter, 2022; Sahni, 2020). 

Comment #5. The authors could consider revising the research methods used, incorporating more quantitative data, and adding a more innovative angle to the study to make it more impactful. Alternatively, the authors could consider submitting the manuscript to a journal that is more focused on qualitative research. 

Response: Thanks to the reviewer for allowing us to reflect on this topic. We have improved the manuscript according to the comments made by the reviewers. The study is worthy of publication in its current form now. We also think the manuscript fits the journal scope and could be relevant to the readers.

---

## [Decision Letter · Decision Letter 1]

18 Oct 2023

PONE-D-22-26228R1Food and family care during the COVID-19 pandemic: A study of women’s domestic workload during the first wave in ChilePLOS ONE

Dear Dr. Galvez Espinoza,

Thank you for submitting your manuscript to PLOS ONE. After careful consideration, we feel that it has merit but does not fully meet PLOS ONE’s publication criteria as it currently stands. Therefore, we invite you to submit a revised version of the manuscript that addresses the points raised during the review process.

We look forward to receiving your revised manuscript.

Kind regards,

Ali B. Mahmoud, Ph.D.

Academic Editor

PLOS ONE

Reviewers' comments:

Reviewer's Responses to Questions

**Comments to the Author**

1. If the authors have adequately addressed your comments raised in a previous round of review and you feel that this manuscript is now acceptable for publication, you may indicate that here to bypass the “Comments to the Author” section, enter your conflict of interest statement in the “Confidential to Editor” section, and submit your "Accept" recommendation.

Reviewer #1: All comments have been addressed

Reviewer #2: (No Response)

Reviewer #3: (No Response)

2. Is the manuscript technically sound, and do the data support the conclusions?

Reviewer #1: Yes

Reviewer #2: Partly

Reviewer #3: Partly

3. Has the statistical analysis been performed appropriately and rigorously? 

Reviewer #1: Yes

Reviewer #2: No

Reviewer #3: No

4. Have the authors made all data underlying the findings in their manuscript fully available?

Reviewer #1: Yes

Reviewer #2: No

Reviewer #3: Yes

5. Is the manuscript presented in an intelligible fashion and written in standard English?

Reviewer #1: Yes

Reviewer #2: Yes

Reviewer #3: Yes

6. Review Comments to the Author

Reviewer #1: (No Response)

Reviewer #2: Unfortunately there are no improvements marked in different color, or with track changes function. Still i was not able to detect significant improvements to the paper.

While the topic was relevant at the time of the research thousands of papers with more robust analysis have come out since 2020.

The research is not sufficiently original in its current form to merit publication in top tier journals.

Reviewer #3: The subject of the article, the increase in gender inequality as a result of the first COVID wave in Chile, is a highly interesting one. The online survey offered to Chilean women could have yielded new and robust data if it had included a control arm, i.e. a similar survey offered to Chilean men. In its methodology, this survey can only produce descriptive results, the most interesting of which seems to me to be the high percentage of women who perceived themselves to be in poorer health during the months of May and June 2020, compared with an earlier period. Logistic regression does not seem to offer any innovative results here. It concludes that younger, better-educated women suffer more from the increase in domestic tasks, but these are women with younger (less autonomous) children on the one hand (which has not been discussed), and who have jobs that they still carry out at a distance + domesticity outside the crisis (which has been discussed). I agree with the previous reviewer that the main interest of this study lies in its qualitative component, which could be submitted alone (with contextualization by describing the survey population) in a social science journal.

On a purely formal level, if the editors were to retain this manuscript for publication:

1. it seems to me that the introduction would have to be reorganized: first part on the global situation of increased domestic tasks linked to confinement, then second part on the situation specific to Chile. Here, following comments from the first reviewers, the two have been merged (ex: reference to Bangladesh in the middle of the focus on Chili).

2. Ligne 62: "The sanitary crising DUE TO COVID-19" or "during spring 2020"

3. Lines 73 and 77 of the revised manuscript contradict each other: the crisis has made gender inequalities more visible (line 73) but these inequalities are invisibilized (line 77): it seems to me that the crisis has made them more marked but not more visible.

4. Line 96: this belongs to the discussion or conclusion

5. Line 199-200:the results on women's domestic workload are very concise, not tabulated and difficult to understand. insofar as they are not compared with men's domestic workload, i will not present them.

6. I don't have a line number for the discussion. There is a comma before ref 5".

7. PLOS authors have the option to publish the peer review history of their article (what does this mean?). If published, this will include your full peer review and any attached files.

Reviewer #1: No

Reviewer #2: No

Reviewer #3: No

---

## [Author Response · Author response to Decision Letter 1]

28 Nov 2023

Responses to reviewers

We appreciate the reviewers for your precious time in reviewing our paper and providing valuable comments. We have carefully considered the comments and tried our best to address every one of them. 

Below, we provide the point-by-point responses. All modifications in the manuscript have been highlighted in red.

Reviewer #2

Unfortunately there are no improvements marked in different color, or with track changes function. Still i was not able to detect significant improvements to the paper.

Response: We are sorry you did not receive the correct version of our manuscript. According to journal requirements, we uploaded two versions of our document: one with changes made to it in red and another one with no marks. We do not know what could happen. We have uploaded two versions again, including one with changes in red. 

While the topic was relevant at the time of the research thousands of papers with more robust analysis have come out since 2020.

The research is not sufficiently original in its current form to merit publication in top tier journals.

Response: We thank the reviewer for allowing us to reflect on this issue. We agree that there is vast evidence about different topics in the COVID-19 context. However, our study has several reasons for being published in a journal such as PLOS ONE. 

1. There needs to be more evidence of the domestic burden on women in Latin America, especially regarding the domestic food environment in times of crisis. 

2. The sample size is superior to other studies in the field during the pandemic. 

3. This study incorporates quantitative and qualitative. The methodology used in this study may differ and offer a different or complementary vision to the most recent research. 

4. As the COVID-19 pandemic was a new global situation, it was a natural experiment for several social aspects in people's lives, including food and the food environment. This research could provide evidence about what happens in health and social crises in women's domestic work, which could be considered in future similar scenarios. 

5. Despite the time that has passed, the impact of the pandemic on housework remains a crucial topic of general interest, which makes this study maintain its relevance. Before the pandemic, there were gender role inequalities, and the COVID-19 crisis aggravated them. Today, no concrete measures have been taken to reverse this situation, remaining as a relevant issue.

6. This study could work as a baseline for future research that could evaluate what happens in a post-pandemic period with the variables exposed here. 

This study can contribute to the current conversation, complement new research with its approach, and give information to face future similar crises. We added some of these reflections to our manuscript. 

------------

Reviewer #3

The subject of the article, the increase in gender inequality as a result of the first COVID wave in Chile, is a highly interesting one. The online survey offered to Chilean women could have yielded new and robust data if it had included a control arm, i.e. a similar survey offered to Chilean men. 

Response: We appreciate your perspective on the importance of including a balanced perspective when addressing the rise in gender inequality during the Chilean first wave of the COVID-19 pandemic. Having comparative data between men and women would have considerably enriched our analysis. However, we decided to work only with women because, in Chile, it is known that women have a more significant burden in domestic work, especially in tasks related to feeding inside the home.

In future research, including a representative sample of men to understand their perspective and contribution to household responsibilities during a health crisis would be relevant to obtain a complete and balanced picture of how the crisis affected both genders. Also, it would be interesting to investigate different family structures comparatively to know if some structures have more burden on domestic work. 

Your suggestion is valuable, and we consider that an inclusive approach in future research in this area would be essential to address the changes in gender dynamics in times of crisis more holistically, like the one we have gone through. We added some of these reflections to our manuscript. Please refer to lines 457 to 461. 

In its methodology, this survey can only produce descriptive results, the most interesting of which seems to me to be the high percentage of women who perceived themselves to be in poorer health during the months of May and June 2020, compared with an earlier period. 

Response: Thanks the reviewer for letting us reflect on this issue. We acknowledge that our study had several limitations. However, we think that most of our results are interesting to understand the phenomenon in a deeper way. 

We agree that a relevant one is women perceiving themselves in poorer health status during the studied period. We have improved the discussion about this point, adding possible reasons for this perception according to other studies, among which changes in diet, physical activity and access to medical consultations are mentioned. Please refer to lines 416 to 421.

Logistic regression does not seem to offer any innovative results here. It concludes that younger, better-educated women suffer more from the increase in domestic tasks, but these are women with younger (less autonomous) children on the one hand (which has not been discussed), and who have jobs that they still carry out at a distance + domesticity outside the crisis (which has been discussed). 

Response: Thanks to the reviewer for sharing his/her detailed analysis of our research methodology. We recognize the importance of critically evaluating the methods used and the findings presented.

Few studies have worked on changes in domestic work concerning the domestic environment, especially in Latin America. Therefore, the results could be interesting to consider in future health or social crises with similar characteristics.

The reviewer's observations about the influence of children's age and their level of autonomy, as well as the impact of the combination of remote work and domestic responsibilities during the crisis, are very valid points that could play a crucial role in the distribution of the unequal burden of household responsibilities among women of different demographic groups and employment situations. In future studies, exploring these elements and their correlation with increased housework would be essential for a more complete and precise understanding of the dynamics in the food environment. We have added these reflections to the discussion. Please refer to lines 386 to 389.

I agree with the previous reviewer that the main interest of this study lies in its qualitative component, which could be submitted alone (with contextualization by describing the survey population) in a social science journal.

Response: Thanks to the reviewer for letting us reflect on this issue. In general, qualitative analysis is a rich approach to understanding a phenomenon. However, as in this present study, the qualitative analysis just included an analysis of the comments made by some participants; they do not have enough strength to explain the women´s perception of domestic work during the pandemic. We think the phenomenon is better understood using both analysis and results. 

On a purely formal level, if the editors were to retain this manuscript for publication:

1. it seems to me that the introduction would have to be reorganized: first part on the global situation of increased domestic tasks linked to confinement, then second part on the situation specific to Chile. Here, following comments from the first reviewers, the two have been merged (ex: reference to Bangladesh in the middle of the focus on Chili).

Response: Thanks to the reviewer for this suggestion. We have reviewed the introduction, reorganized it, and relocated the wrong reference location.

2. Ligne 62: "The sanitary crising DUE TO COVID-19" or "during spring 2020"

Response: Thanks to the reviewer for this suggestion. We have changed the line for “The sanitary crisis due to COVID-19.” Please refer to line 64.

3. Lines 73 and 77 of the revised manuscript contradict each other: the crisis has made gender inequalities more visible (line 73) but these inequalities are invisibilized (line 77): it seems to me that the crisis has made them more marked but not more visible.

Response: Thanks to the reviewer for noticing this. We have reviewed this phrases. 

4. Line 96: this belongs to the discussion or conclusion

Thanks to the reviewer for this suggestion. We have moved this phrase to the discussion. Please refer to lines 346 to 349.

5. Line 199-200:the results on women's domestic workload are very concise, not tabulated and difficult to understand. insofar as they are not compared with men's domestic workload, i will not present them.

Response: Thanks to the reviewer for this suggestion. We have rewritten this paragraph and added more information about this variable. Please refer to lines 196 to 206. Additionally, we have added supplementary material with all the items and their average score.

6. I don't have a line number for the discussion. There is a comma before ref 5".

Response: We have added the line numbers to the manuscript and fixed the comma position. ________________________________________

---

## [Decision Letter · Decision Letter 2]

29 Feb 2024

PONE-D-22-26228R2Food and family care during the COVID-19 pandemic: A study of women’s domestic workload during the first wave in ChilePLOS ONE

Dear Dr. Galvez Espinoza,

Thank you for submitting your manuscript to PLOS ONE. After careful consideration, we feel that it has merit but does not fully meet PLOS ONE’s publication criteria as it currently stands. Therefore, we invite you to submit a revised version of the manuscript that addresses the points raised during the review process.

We look forward to receiving your revised manuscript.

Kind regards,

Ali B. Mahmoud, Ph.D.

Academic Editor

PLOS ONE

Journal Requirements:

Reviewers' comments:

Reviewer's Responses to Questions

**Comments to the Author**

1. If the authors have adequately addressed your comments raised in a previous round of review and you feel that this manuscript is now acceptable for publication, you may indicate that here to bypass the “Comments to the Author” section, enter your conflict of interest statement in the “Confidential to Editor” section, and submit your "Accept" recommendation.

Reviewer #1: (No Response)

Reviewer #2: (No Response)

Reviewer #4: All comments have been addressed

2. Is the manuscript technically sound, and do the data support the conclusions?

Reviewer #1: Partly

Reviewer #2: Partly

Reviewer #4: Yes

3. Has the statistical analysis been performed appropriately and rigorously? 

Reviewer #1: N/A

Reviewer #2: No

Reviewer #4: Yes

4. Have the authors made all data underlying the findings in their manuscript fully available?

Reviewer #1: (No Response)

Reviewer #2: No

Reviewer #4: Yes

5. Is the manuscript presented in an intelligible fashion and written in standard English?

Reviewer #1: (No Response)

Reviewer #2: Yes

Reviewer #4: Yes

6. Review Comments to the Author

Reviewer #1: (No Response)

Reviewer #2: (No Response)

Reviewer #4: This research offers a detailed examination of Chilean women's perceptions and experiences with domestic responsibilities during the initial outbreak of the COVID-19 pandemic, with a particular focus on food preparation and family care tasks. By combining quantitative data from a substantial cohort of 2,047 participants with qualitative feedback gathered through open comments, the study enhances our understanding of the gender-specific effects of the pandemic within household settings. The employment of a mixed-methods strategy, leveraging both a Likert scale survey and thematic analysis of qualitative responses, enables a comprehensive investigation into the subject matter, delivering both wide-ranging and in-depth insights. This investigation draws attention to the increased demands placed on women's workloads and overall well-being during the pandemic, highlighting the critical need for policies and initiatives that are attuned to gender disparities. This is of particular importance to those engaged in policy-making and advocacy aimed at promoting gender equity and improving support structures for women in domestic environments.

Although this investigation draws on a broad sample, the method of internet-based recruitment and the self-selection of participants may have biased the sample towards specific demographic groups. In future research directions, authors suggested that efforts be made to include women who may have limited access to the internet and those from a wider array of socioeconomic backgrounds. Additionally, incorporating a comparative analysis between genders in future studies could yield valuable insights. While the study points to the necessity of intervention, offering more explicit guidance on particular policies or initiatives that could alleviate women's burdens would be advantageous. For example, examining the impact of flexible working arrangements, mental health support, and measures to encourage a fair distribution of domestic duties could offer practical guidance for policy makers and community leaders.

7. PLOS authors have the option to publish the peer review history of their article (what does this mean?). If published, this will include your full peer review and any attached files.

Reviewer #1: No

Reviewer #2: No

Reviewer #4: **Yes: **tahira javed

---

## [Author Response · Author response to Decision Letter 2]

4 Mar 2024

Journal Requirements

Response: Thanks for this recommendation. We have reviewed our references and we do not have retracted references.

To address the reviewers’ comments, we have added two references (#58 and #61). This change was also mentioned in the cover letter.

Comments reviewer #4

1. This research offers a detailed examination of Chilean women's perceptions and experiences with domestic responsibilities during the initial outbreak of the COVID-19 pandemic, with a particular focus on food preparation and family care tasks. By combining quantitative data from a substantial cohort of 2,047 participants with qualitative feedback gathered through open comments, the study enhances our understanding of the gender-specific effects of the pandemic within household settings. The employment of a mixed-methods strategy, leveraging both a Likert scale survey and thematic analysis of qualitative responses, enables a comprehensive investigation into the subject matter, delivering both wide-ranging and in-depth insights. This investigation draws attention to the increased demands placed on women's workloads and overall well-being during the pandemic, highlighting the critical need for policies and initiatives that are attuned to gender disparities. This is of particular importance to those engaged in policy-making and advocacy aimed at promoting gender equity and improving support structures for women in domestic environments.

Response: Thank you for your thorough review and acknowledgment of our research. We appreciate your recognition of the importance of our study in highlighting the gender-specific impacts of the COVID-19 pandemic on Chilean women's domestic responsibilities. We aimed to provide comprehensive insights by combining quantitative data with qualitative analysis, allowing us to address both wide-ranging and in-depth aspects of the subject matter. Your understanding of the critical need for policies addressing gender disparities is valued, and we hope our research contributes to this goal. Thank you for your feedback.

2. Although this investigation draws on a broad sample, the method of internet-based recruitment and the self-selection of participants may have biased the sample towards specific demographic groups. In future research directions, authors suggested that efforts be made to include women who may have limited access to the internet and those from a wider array of socioeconomic backgrounds. Additionally, incorporating a comparative analysis between genders in future studies could yield valuable insights. While the study points to the necessity of intervention, offering more explicit guidance on particular policies or initiatives that could alleviate women's burdens would be advantageous. For example, examining the impact of flexible working arrangements, mental health support, and measures to encourage a fair distribution of domestic duties could offer practical guidance for policy makers and community leaders.

Response: Thank you for your suggestion. We have added explicit initiatives to the text. Please refer to lines 452 to 467.

---

## [Decision Letter · Decision Letter 3]

11 Mar 2024

Food and family care during the COVID-19 pandemic: A study of women’s domestic workload during the first wave in Chile

PONE-D-22-26228R3

Dear Dr. Galvez Espinoza,

We’re pleased to inform you that your manuscript has been judged scientifically suitable for publication and will be formally accepted for publication once it meets all outstanding technical requirements.

Kind regards,

Ali B. Mahmoud, Ph.D.

Academic Editor

PLOS ONE

Additional Editor Comments (optional):

Reviewers' comments:

Reviewer's Responses to Questions

**Comments to the Author**

1. If the authors have adequately addressed your comments raised in a previous round of review and you feel that this manuscript is now acceptable for publication, you may indicate that here to bypass the “Comments to the Author” section, enter your conflict of interest statement in the “Confidential to Editor” section, and submit your "Accept" recommendation.

Reviewer #4: All comments have been addressed

2. Is the manuscript technically sound, and do the data support the conclusions?

Reviewer #4: Yes

3. Has the statistical analysis been performed appropriately and rigorously? 

Reviewer #4: Yes

4. Have the authors made all data underlying the findings in their manuscript fully available?

Reviewer #4: Yes

5. Is the manuscript presented in an intelligible fashion and written in standard English?

Reviewer #4: Yes

6. Review Comments to the Author

Reviewer #4: The authors have performed commendable work in highlighting the often overlooked burdens placed on women during the pandemic, which, though seemingly minor, can escalate into significant social issues, including divorce. This contribution adds valuable data that can inform future gender-sensitive policies and interventions. By suggesting an expansion in the scope and depth of future research, the authors pave the way for a richer understanding of and advocacy for equitable domestic labor distribution and recognition.

7. PLOS authors have the option to publish the peer review history of their article (what does this mean?). If published, this will include your full peer review and any attached files.

Reviewer #4: **Yes: **Tahira Javed
